# Expansion of MK Circle Theory for Dyads and Triads

Sean Mather * and Arthur Erdman

Department of Mechanical Engineering, University of Minnesota, Twin Cities, Minneapolis, MN 55455, USA;
agerdman@umn.edu
* Correspondence: mathe587@umn.edu

**Abstract:** The MK circles represent a kinematic synthesis tool for the dimensional synthesis of planar dyads. The tool is uniquely useful in its ability to both find specific dyad solutions and help the designer visualize numerous potential dyad pivot locations in the solution space. Here, the existing understanding of MK circles is summarized for three and four specified motion positions and extended for additional positions. Then the technique is expanded to show its application to MKT circles for triad synthesis, including solution space visualization, ground pivot specification, and multi-loop synthesis of complex mechanisms. These methods are illustrated by a unifying example that provides a sample procedure for applying the MK/MKT circles, and implements each of the aforementioned techniques. The interchangeability of loop-based synthesis approaches is demonstrated by comparing the new methodology to the compatibility linkages technique.

**Keywords:** kinematic synthesis; MK circle; dyad; triad; planar mechanisms





## 1. Introduction

Dyads and triads are two- and three-link chains that may be viewed as the building blocks for planar linkage mechanisms. The MK[1] [1–3] circles are a useful tool for both finding solutions to mechanism design problems and for visualizing properties of the possible pivot location solution space. The name MK circle stems from the German words, "Mittelpunkt" and "Kreispunkt", which refer to the center point (ground pivot) and circle point (moving pivot) of a dyad. This information is compiled and updated in a new way, and then expanded to show its applications to triads in several prescribed positions. The property of kinematic synthesis that dyad pivot location solutions settle into circles was first observed by Loerch, who demonstrated the existence of these circles, identified some of their properties, and found solutions for up to five prescribed positions of path generation [1]. Mlinar expanded on these findings to include triad synthesis, but primarily focused on the existence of forbidden regions in triad synthesis and did not lean into the potential implications for identifying solution triads [3,4].

This paper provides an improved understanding of synthesis of dyads and triads using MK circles by encapsulating the key ideas of both previous works and expanding upon them. Here, we demonstrate the effectiveness of the synthesis method for motion problems in five prescribed positions for a dyad and up to seven positions for a triad. A geometry-inspired algorithm is developed for finding circle intersections in terms of the free-choice variable. Further, the method is demonstrated to be effective for specifying the ground pivot location of a triad chain for multi-loop synthesis, an approach which was not previously considered. Finally, a practical example is provided which shows how designers may apply the multi-loop synthesis approach advocated in this paper to real problems. This example unifies the theory and shows how it may be practically useful, a concern which was not addressed in the previous works. Additionally, the example proves that other loop-based synthesis approaches for deriving dyad and triad chains are compatible with the present method [2,5–10]. While the MK circles are effective for finding dyad and triad solutions using either R or P joints, all figures included in the present work show solutions comprised of pinned (R) $f_1$ joints.

## 2. Body

When performing the dimensional synthesis of a dyad (see Figure A2a), it is possible to find a solution for the vectors representing this dyad (and therefore the pivot locations in their first position) directly through linear algebra for up to three positions, using the "standard form" equations shown in Equation (1). This matrix equation is generated by vector loops–Figure A2a yields the first row in this equation.

$$\begin{bmatrix} e^{i\beta_2} - 1 & e^{i\alpha_2} - 1 \\ e^{i\beta_3} - 1 & e^{i\alpha_3} - 1 \end{bmatrix} \begin{bmatrix} W \\ Z \end{bmatrix} = \begin{bmatrix} \delta_2 \\ \delta_3 \end{bmatrix} \tag{1}$$

In three positions of motion generation, $\delta_2$, $\alpha_2$, $\delta_3$, and $\alpha_3$ are prescribed and both the angles $\beta_2$ and $\beta_3$ are free choices. If a designer holds the value of $\beta_2$ constant in Equation (1), then iterates through many values of $\beta_3$, plotting the XY coordinates of either the ground pivot (vector origin for W) or the moving pivot (vector origin for Z) will cause a circle to emerge. Each point on this circle represents a unique pivot location which is a solution to Equation (1). The standard form equations for a dyad in four positions are included in Appendix A.

A designer may choose any of these pivot locations to generate one dyad, effectively choosing the value of $\beta_3$. If no satisfactory solutions exist on the circle (for example, an acceptable ground or moving pivot location), they may choose a new value of $\beta_2$, which then generates a new circle of $\beta_3$ values. One such pair of MK circles is seen in Figure 1. Consistently throughout this paper, figures use red points to indicate ground pivot locations, and blue points to indicate moving pivot locations.

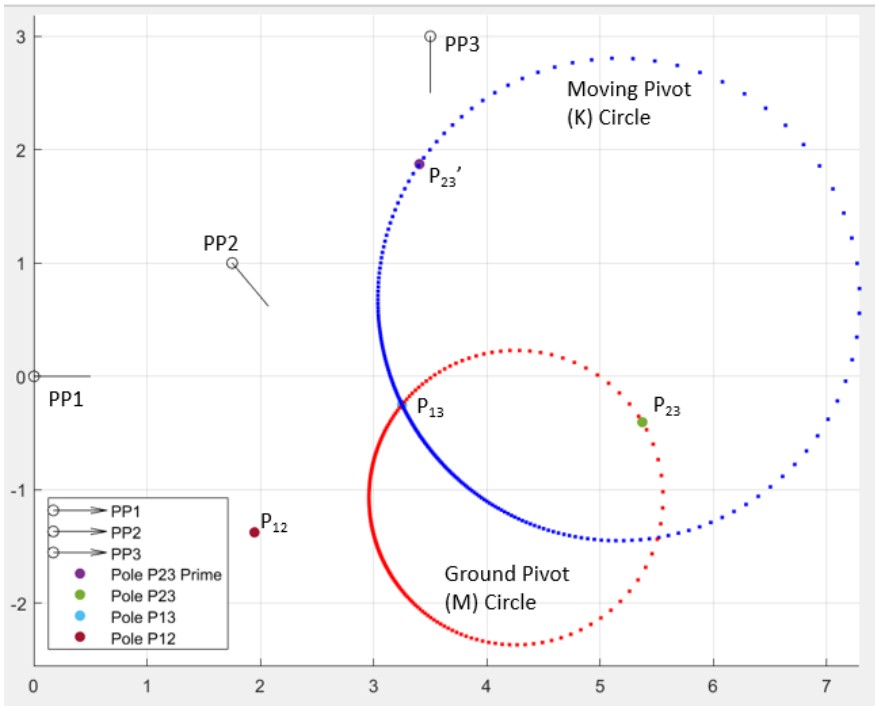

**Figure 1.** A depiction of the MK circles for a dyad in three positions. PP1 = (0, 0). $\delta_2$ = 1.75 + 1i. $\delta_3$ = 3.5 + 3i. $\alpha_2$ = −50, $\alpha_3$ = −90, and $\beta_2$ = −110. The red circle represents potential ground pivot locations, while the blue circle represents potential moving pivot locations.

## 3. Poles

Poles are linked to the MK theory in the same way the poles were a critical part of Burmester's original planar synthesis theory [11–13]. The pole positions are also shown in Figure 1 along with the MK circles, and they denote special points in the plane; given any two prescribed positions of a moving plane including their rotation angles, the pole

represents the fixed pivot location about which a rigid link would purely rotate that plane from one position to the other. For dyads, the location of poles of the moving plane can be precisely found using Equations (2)–(4), which express the position of the pole relative to the first prescribed position of the moving plane [2] (p. 119). The image poles express a similar concept but are found by reflecting their corresponding natural pole across a line formed by two other poles. For example, in Figure 1, pole $P_{23'}$ is found by reflecting the point $P_{23}$ across the line $\overline{P_{12}P_{13}}$.

In Figure 1, it is observed that although a uniform increment is used for each value of $\beta_3$, the solution density of ground and moving pivots is highest around certain poles, and the density is the lowest at the opposite end of the circle. Notice that for the ground pivot circle of a three-position problem, this high-density region forms around pole $P_{13}$. While the circles are continuous if an infinite number of points are plotted, this indicates that a larger range of $\beta_3$ values have their solution adjacent to the given pole.

$$\mathbf{P_{12}} = \frac{\delta_2}{1 - e^{i\alpha_2}} \tag{2}$$

$$\mathbf{P_{23}} = \frac{\delta_3 e^{i\alpha_2} - \delta_2 e^{i\alpha_3}}{e^{i\alpha_2} - e^{i\alpha_3}} \tag{3}$$

Other poles follow the same equation patterns, with any pole relative to position one using Equation (2) (with the appropriate change in numbers, e.g., substitute subscript 3's for 2's), and poles between any other two positions are found using Equation (3). Image poles are found by first identifying the base pole (i.e., $P_{23}$ to find $P_{23'}$), and then using Equation (4).

$$\mathbf{P'_{23}} = 2w - \text{real}(P_{23}) + i(2dm - \text{imag}(P_{23}) + 2b) \tag{4}$$

In which b is the y-intercept coordinate of the line passing through the points $P_{12}$ and $P_{23}$, m is the slope of that same line, and d is the expression shown in Equation (5). Real() and imag() denote the real and imaginary components of the pole [14].

$$\mathbf{w} = \frac{\mathbf{real}(\mathbf{P_{23}}) + (\mathbf{imag}(\mathbf{P_{23}}) - \mathbf{b})\mathbf{m}}{1 + \mathbf{m}^2} \tag{5}$$

Poles have special significance in the dyad solution space. In Figure 1, the value of free choice $\beta_2$ has been held constant, while $\beta_3$ was varied. If $\beta_2$ is also varied, an interesting pattern emerges. In Figure 2, six values of $\beta_2$ are plotted with the same several hundred values of $\beta_3$ as before. Each unique value of $\beta_2$ generates a new circle. What makes this outcome so interesting, though, is that each of these circles passes through the same two points, which happen to be the poles.

These properties observed for the MK circles are due to each pole representing a special case of the selected free choices that will be present on every circle. For example, $P_{13}$ is a consistent ground pivot solution because of the case when $\beta_3 = \alpha_3$. In this instance, the dyad will behave similarly to a single link as it rotates from position one to three, and therefore the only place a ground pivot could exist is $P_{13}$. Similarly, $P_{13}$ is a consistent moving pivot location because of the special case where $\beta_3 = 0$, in which case all motion between positions one and three would have to stem from the rotation of the moving plane about the pivot point $K_1$. The poles are also useful as they mark points that the Burmester curves pass through for the four specified positions, allowing for a rough sketch of the shape of the curves with limited calculation [2,15].

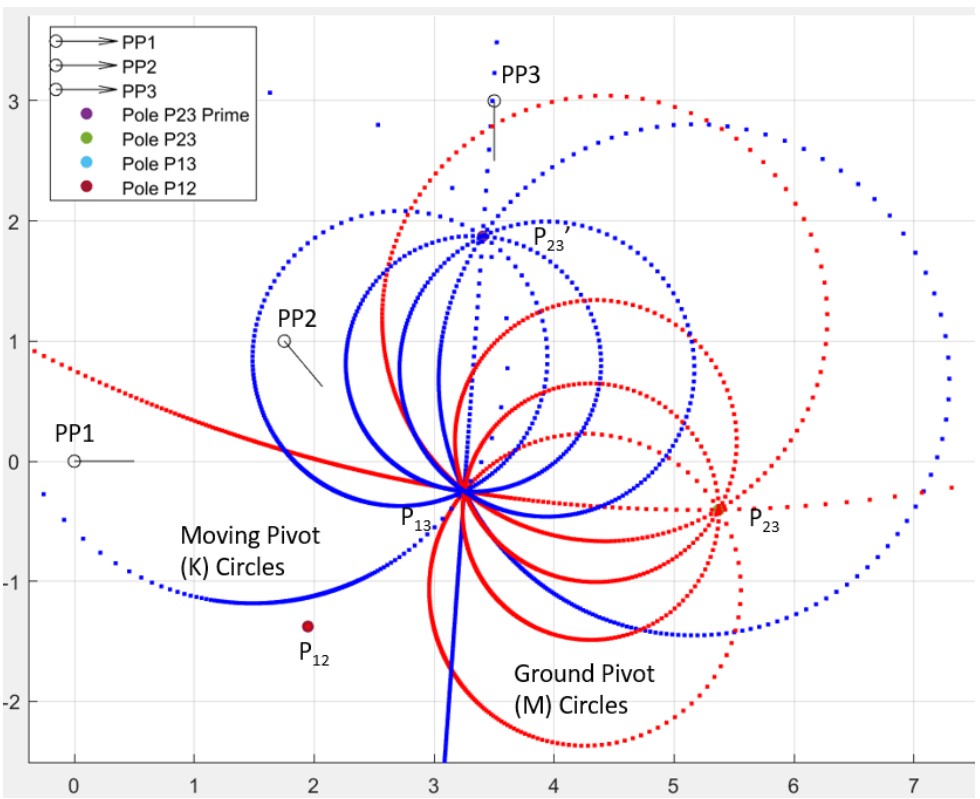

**Figure 2.** Six values of $\beta_2$ are plotted with several hundred values of $\beta_3$ each. $\beta_2$ has an initial value of $-110$ degrees, as in Figure 1, and the increment between each value of $\beta_2$ is 60 degrees.

At times it may be valuable to create the circles directly. In this case, there are two primary approaches to finding the center points and radius of the circles. Any circle may be defined by three points which it intersects, so one process is to find three points on each circle and then derive the circle description from these points. The other approach is purely geometric and is derived from the pole positions: for three positions, find each of the natural poles and the image pole $P_{23'}$ using Equations (2)–(4). $P_{13}$ and $P_{23}$ are associated with the M circle, and $P_{13}$ and $P_{23'}$ are associated with the K circle. To find the circle centers, draw a perpendicular bisector through each of these two pairs of poles. These two lines are the centerlines on which the center points of each circle will lie. For each unique value of $\beta_2$, use the angular relationships established in Equations (6) and (7) to find the center points ($C_M$ and $C_K$) of the M and K circles, respectively.

$$\angle P_{13}C_M P_{23} = \beta_2 = 2\theta_M \tag{6}$$

$$\angle P_{13}C_K P_{23}' = \alpha_2 - \beta_2 = 2\theta_k \tag{7}$$

These angles locate the position of the center of each circle. Once located, the radius of each circle is given in Equations (8) and (9). For the M circle, if the value of $\beta_2$ is negative, an observer viewing pole $P_{13}$ from the center of the circle would rotate clockwise by the angle $|\beta_2|$ to arrive at pole $P_{23}$ (as in Figure 3). If the value of $\beta_2$ is positive, the observer rotates counterclockwise from $P_{13}$ to $P_{23}$.

$$r_M = \overline{P_{13}C_M} = \frac{d/2}{\sin(\theta_M)} \tag{8}$$

$$r_K = \overline{P_{13}C_K} = \frac{d/2}{\sin(\theta_K)} \tag{9}$$

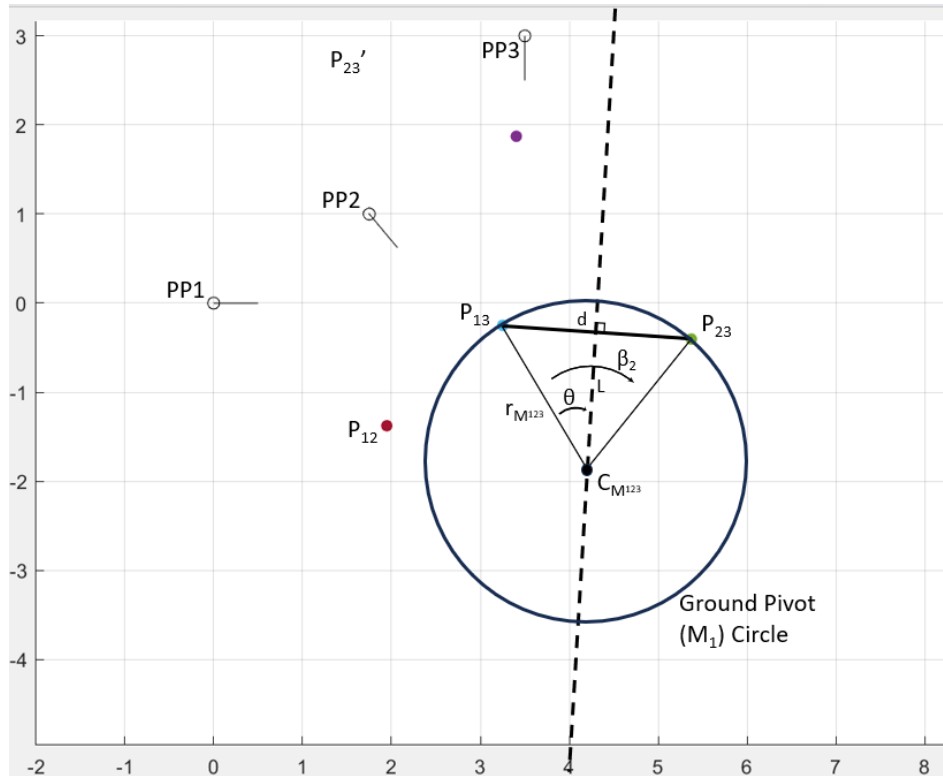

**Figure 3.** A depiction of a single ground pivot circle with the key geometric parameters labeled for the identification of the center point coordinates and radius of the circle in terms of $\beta_2$.

Using this approach, the complete set of dyad solutions for a particular value of $\beta_2$ is found without using the standard form equation.

If the motion generation problem requires four precision positions, a linear solution is no longer possible, which at first glance seems to render the MK circles useless, as plotting them relies on a linear solver technique. However, there still is a practical application for them. As an example, consider first finding a possible ground pivot (M location) for the four-position problem; it is impossible to make a single circle that spans the four positions, but the solution can be reimagined as an intersection of two sets of M circles, with each set drawn from three of the four positions. Typically, the first circle is taken from precision positions 1, 2, and 3, while the second is created using positions 1, 2, and 4. Where these circles intersect represents a pivot location that fulfills both the first three positions, and positions 1, 2, and 4. For any given value of $\beta_2$, iterating through values of $\beta_3$ will generate the 123 circle, and iterating through values of $\beta_4$ will generate the 124 circle. The pairs of circles will intersect zero, one (if the circles are tangent to one another), or two times, indicating the number of solutions available for the given value of $\beta_2$. These intersecting circle pairs for both M and K circles can be seen in Figure 4.

For a motion-generation dyad, the maximum number of prescribed positions with an exact solution is five (See Table 1). There are no free choices in this case. An example figure of the MK circles for a dyad in five prescribed positions is shown in Appendix B. The geometric formulation introduced in Equations (2)–(9) allows for the formulation of a new algorithm which may be used to identify intersections between these sets of circles in terms of the free choice variable $\beta_2$. The procedure is as follows.

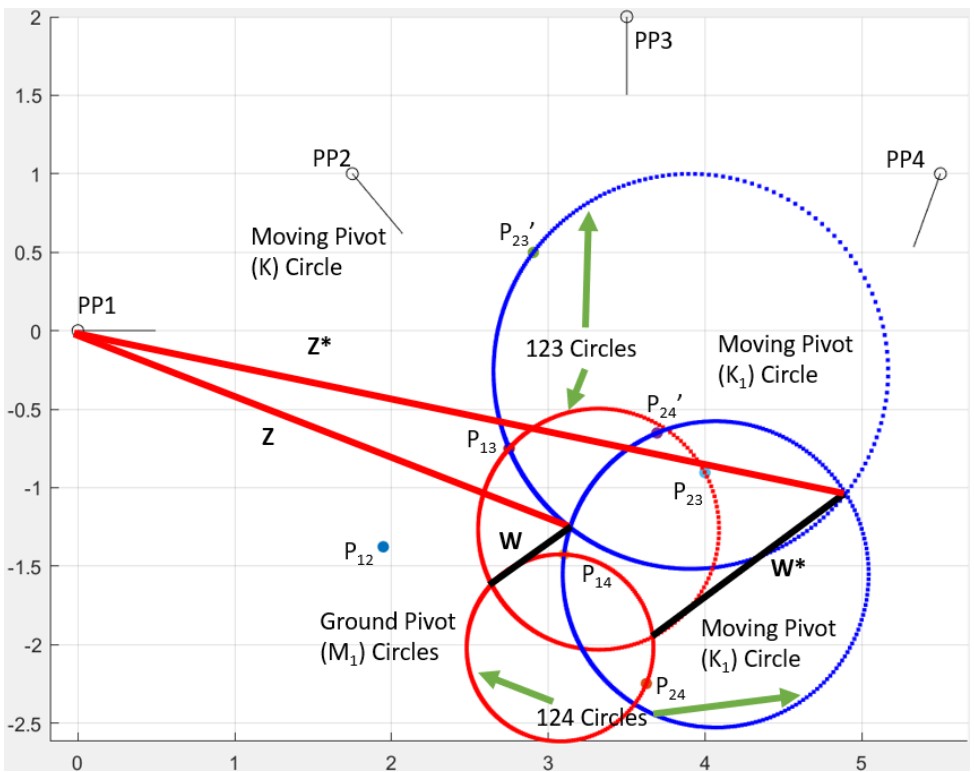

**Figure 4.** Sample view of the MK circles for a dyad in four precision positions, with the poles, **W**, and **Z** vectors shown. Here, PP1 = 0 + 0i, PP2 = 1.75 + 1i, PP3 = 3.5 + 2i, PP4 = 5.5 + 1i. $\alpha_2 = -50$, $\alpha_3 = -90$, $\alpha_4 = -110$. Pictured circles are for $\beta_2 = -110$. **W** = 0.514 + 0.382i, **Z** = −3.141 + 1.245i. **W*** = 1.232 + 0.894i, **Z*** = −4.895 + 1.036i.

**Table 1.** A classification of the prescribed data and maximum number of potential solutions for each number of prescribed positions of a dyad.

| Number of Positions | Prescribed Values | Unknowns | Free Choices | Number of Solutions |
|---|---|---|---|---|
| 2 | $\delta_2, \alpha_2$ | 5 *—**W**, **Z**, $\beta_2$ | 3—$\beta_2$, **W** or **Z** | $\infty^3$ |
| 3 | $\delta_{2-3}, \alpha_{2-3}$ | 6—**W**, **Z**, $\beta_{2-3}$ | 2—$\beta_{2-3}$ | $\infty^2$ |
| 4 | $\delta_{2-4}, \alpha_{2-4}$ | 7—**W**, **Z**, $\beta_{2-4}$ | 1—$\beta_2$ | $\infty^1$ |
| 5 | $\delta_{2-5}, \alpha_{2-5}$ | 8—**W**, **Z**, $\beta_{2-5}$ | 0 | Finite |

* Because **W** and **Z** are both vector quantities, they account for two unknowns each. Table shows the number of free choices available for a dyad based on the number of prescribed positions. Notice that for three positions (Equation (1)) there are two free choices, which are represented by the MK circles as $\beta_2$ and $\beta_3$ are varied. For the cases of four and five positions, this paper shows how multiple sets of MK circles are combined to yield dyad solutions.

First, find the coordinates and radius of each circle in terms of $\beta_2$ using Equations (6)–(9). If the inter-center distance between any two circles is greater than $r_{M1} + r_{M2}$, the circles do not intersect, and no solutions exist for that given value of $\beta_2$. Second, the distance from each circle center to the radical line must be found. For any two non-congruent circles, the radical line is the line passing through their intersection points [16,17]. These distances are given in Equation (10), where d is the distance between the circle's center-points.

$$a = \frac{r_{M1}^2 - r_{M2}^2 + d^2}{2d} \tag{10}$$

Using this term, a, the distance along the radical line between the intersection point and the line between the circle centers is calculated using Equation (11).

$$y_1 = \sqrt{r_1^2 - a^2} \tag{11}$$

With the variables a, d, and $y_1$ determined, all the required information is known to identify the intersection points. The last step is to adjust the coordinate system to align the local system with the global coordinate system. This is achieved by multiplying by unit vectors set in the proper directions. The two vectors are given in Equations (12) and (13) [18].

$$\boldsymbol{e_1} = \frac{1}{d} * \begin{pmatrix} C_{M2x} - C_{M1x} \\ C_{M2y} - C_{M1y} \end{pmatrix} \tag{12}$$

$$\boldsymbol{e_2} = \frac{1}{d} * \begin{pmatrix} 0 & -1 \\ 1 & 0 \end{pmatrix} \tag{13}$$

These vectors are utilized in the following expression, which locates the one or two intersection points between two circles relative to the center of the first circle.

$$Int_{1,2} = C_{M1} + a * \boldsymbol{e_1} \pm y * \boldsymbol{e_2} \tag{14}$$

This function is determined entirely by the value of $\beta_2$ and the pole positions, which are set by the problem definition. The simplest way to implement this expression for the five-position case is to identify the intersection points between the first two circles, and then find the intersections between the third circle and either of the first two. A viable value of $\beta_2$ is determined in any case where a matching intersection point is found.

## 4. Triad MKT Circles

The MK circle concepts can be expanded to apply to triads as well, with many of the core concepts above having parallels in triad synthesis. The shorthand 'MKT' will be used in this paper, with the T representing the third, triad circle. Each point on this circle is the second moving pivot of a triad chain; this can also be thought of as the end of **Z** or the tail of **V** which extends out to the precision point. Figure A2b shows a triad in two positions with the appropriate labeling.

## 5. Triad Circles

One key difference in finding a single circle set for the triad versus the dyad, is the size of the matrices, and consequently, the number of positions considered. The standard form equations of a triad are shown in Equation (15), for four positions.

$$\begin{bmatrix} e^{i\beta_2} - 1 & e^{i\alpha_2} - 1 & e^{i\gamma_2} - 1 \\ e^{i\beta_3} - 1 & e^{i\alpha_3} - 1 & e^{i\gamma_3} - 1 \\ e^{i\beta_4} - 1 & e^{i\alpha_4} - 1 & e^{i\gamma_4} - 1 \end{bmatrix} \begin{bmatrix} \mathbf{W} \\ \mathbf{Z} \\ \mathbf{V} \end{bmatrix} = \begin{bmatrix} \delta_2 \\ \delta_3 \\ \delta_4 \end{bmatrix} \tag{15}$$

See Table 2 for a list of prescribed positions of motion generation of a triad along with the corresponding free choices. This table and Equation (8) reveal that the solution to these standard form equations is linear through four motion generation prescribed positions, where the dyad was only linear in up to three positions. This results in the basic case of the triad MKT circle representing four positions, rather than three, e.g., for the dyad. The MKT circles for a typical triad case are shown in Figure 5.

**Table 2.** A classification of the prescribed data and maximum number of solutions for each number of prescribed positions of a triad.

| Number of Positions | Prescribed Values | Unknowns | Free Choices | Number of Solutions |
|---|---|---|---|---|
| 2 | $\delta_2$, $\alpha_2$, $\gamma_2$ | 7 *—**W**, **Z**, **V**, $\beta_2$ | 5—$\beta_2$, **W** and **Z** | $\infty^5$ |
| 3 | $\delta_{2-3}$, $\alpha_{2-3}$, $\gamma_{2-3}$ | 8—**W**, **Z**, **V**, $\beta_{2-3}$ | 4—$\beta_{2-3}$, **W** | $\infty^4$ |
| 4 | $\delta_{2-4}$, $\alpha_{2-4}$, $\gamma_{2-4}$ | 9—**W**, **Z**, **V**, $\beta_{2-4}$ | 3—$\beta_{2-4}$ | $\infty^3$ |
| 5 | $\delta_{2-5}$, $\alpha_{2-5}$, $\gamma_{2-5}$ | 10—**W**, **Z**, **V**, $\beta_{2-5}$ | 2—$\beta_{2-3}$ | $\infty^2$ |
| 6 | $\delta_{2-6}$, $\alpha_{2-6}$, $\gamma_{2-6}$ | 11—**W**, **Z**, **V**, $\beta_{2-6}$ | 1—$\beta_2$ | $\infty^1$ |
| 7 | $\delta_{2-7}$, $\alpha_{2-7}$, $\gamma_{2-7}$ | 12—**W**, **Z**, **V**, $\beta_{2-7}$ | 0 | Finite |

* Because **W** and **Z** are both vector quantities, they account for two unknowns each.

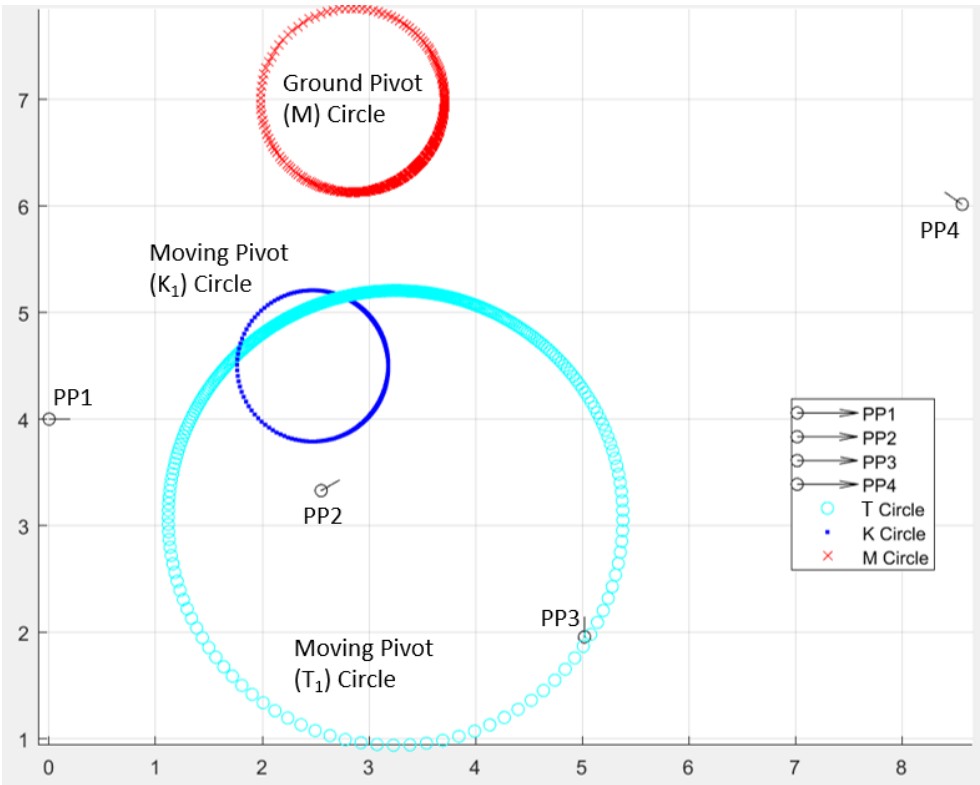

**Figure 5.** A depiction of the MKT circles of a triad in four precision positions. In this example, the $\gamma_j$ and $\beta_j$ angles are prescribed in the problem, indicating the base triad case is a motion generation problem with prescribed timing, and making the $\alpha_j$ angles the free choices. However, any of the three sets of angles can be taken as a free choice by defining the problem differently. Here, PP1 = 0 + 4.0i, PP2 = 2.552 + 3.329i, PP3 = 5.023 + 1.957i, PP4 = 8.564 + 6.014i, $\alpha_2$ = 60. $\gamma_2$ = 30, $\gamma_3$ = 90, $\gamma_4$ = 145, $\beta_2$ = 10, $\beta_3$ = 70, $\beta_4$ = 140.

One additional difference to note is just how many more variables are at play in triad synthesis for the linear solution case (Table 2). There is a whole additional set of rotational angles—the gamma values which represent the angular displacements of the third link—and one more rotational angle in each set (e.g., $\beta_2$, $\beta_3$, and $\beta_4$, as opposed to $\beta_2$ and $\beta_3$ for the dyad). An example of varying multiple free choice variables is shown in Figure 6. In this Figure, $\alpha_2$ and $\alpha_3$ are free choices that locate the center point location of the circles. For each new value of $\alpha_2$ or $\alpha_3$ that is plotted, an additional circle emerges. The points on these pivot circles are found by varying the value of $\alpha_4$ from 0 to 360°. Figure 6 shows six circles of each type (M ground, $K_1$ moving pivot, $T_1$ moving pivot).

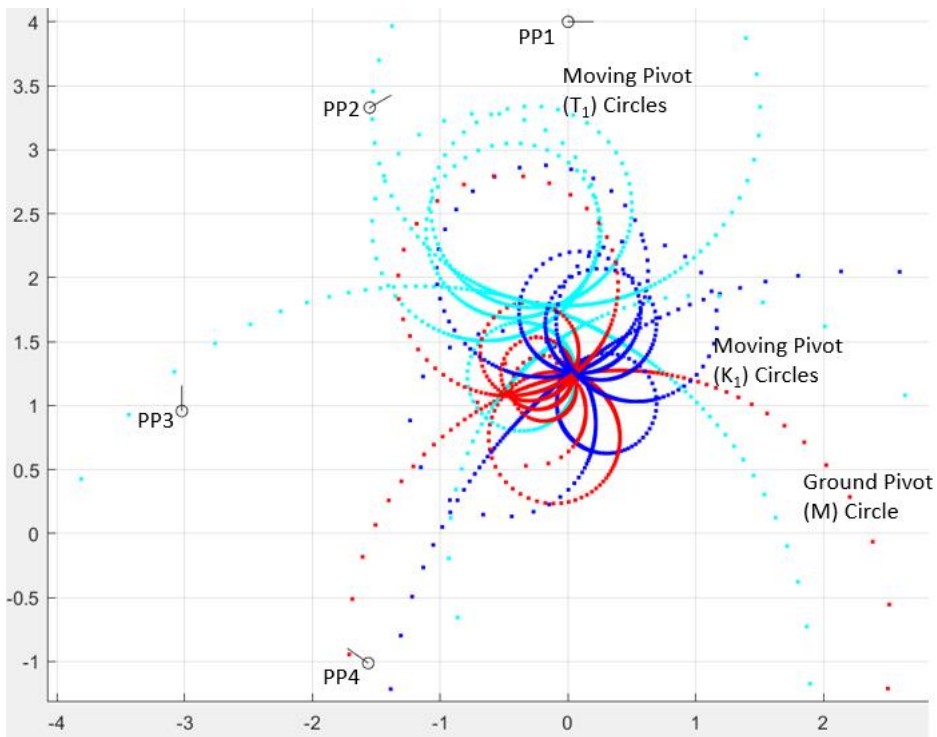

**Figure 6.** A triad in four positions. Testing six values of $\alpha_3$ (resulting in six circles in each M, K and T set) and numerous values of $\alpha_4$ (creating each unique point on the circles). PP1 = 0 + 4i, PP2 = 2.553 + 3.329i, PP3 = 5.023 + 1.957i, PP4 = 8.564 + 6.014i. $\beta_2$ = 60, $\beta_3$ = 70, $\beta_4$ = 140, $\alpha_2$ = 40, $\gamma_2$ = 30, $\gamma_3$ = 90, $\gamma_4$ = 145.

An astute observer may notice that in Figure 6 it appears as though the circles are quite close to portraying the pole behavior visible in Figure 2 for a dyad, with every unique circle in each color-coded set passing through the same two points—revealed in that case to be the poles. However, in this case, there are a few outliers, and not every circle coalesces at the same two points. It seems intuitive that an analogous structure to the dyad poles would exist for the triad, as many similar special cases exist. For instance, if $\beta j = \alpha j$, or $\alpha j = \gamma j$, the triad simplifies to a dyad as Z rotates congruently to W or V, further indicating the existence of special pivot points. A full investigation into the position and properties of these hypothetical triad poles is left for future work, but the concept shows promise for a more complete understanding of the triad solution space.

As with the dyad, the triad can be expanded beyond its linear solution case of four positions: all the way up to seven prescribed positions. Solutions are found by identifying intersections between the respective circles. Figure 7 shows a sample triad for five prescribed positions. Figures depicting the six- and seven-position cases are included in Appendix B.

Although the dyad in five prescribed positions similarly required a three-way intersection and did not allow any free choices, the triad in six prescribed poses does still allow one free choice. Typically, this is chosen as either $\beta_2$ or $\alpha_2$. The solution is found at the three-way intersection of circles formed from positions 1,2,3,4, 1,2,3,5 and 1,2,3,6. A designer may make a free choice for the value of $\beta_2$, then solve for the value of $\beta_3$ which produces a triple intersection point (if indeed any exist, not all values of $\beta_2$ are guaranteed to produce a valid solution) [5] (p. 43). Figure A5 depicts a triple intersection point for a triad solved for six prescribed positions. In Figure A6, a solution triad is shown for a problem defined by seven precision positions. In the seven-position case, four circles must intersect at a single point, and the designer has no free choices. This makes the seven-position case quite difficult to implement practically, as solutions are extremely limited.

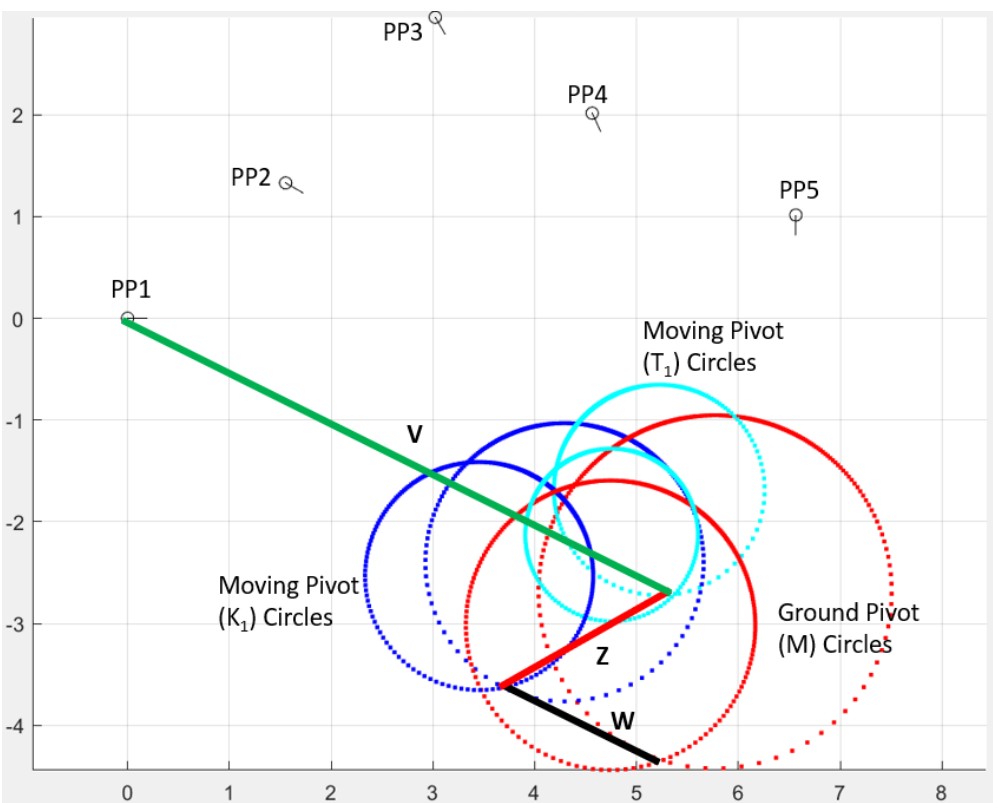

**Figure 7.** The triad MKT circles are shown in five prescribed positions, with one possible solution triad highlighted. PP1 = 0 + 0.0i, PP2 = 1.552 + 1.329i, PP3 = 3.023 + 2.957i, PP4 = 4.564 + 2.014i, PP5 = 6.564 + 1.014i. $\beta_2 = 30$, $\beta_3 = 70$, $\beta_4 = 140$, $\beta_5 = 150$, $\alpha_2 = 60$, $\gamma_2 = -30$, $\gamma_3 = -60$, $\gamma_4 = -65$, $\gamma_5 = -90$. The highlighted solution is **W** = −1.634 + 0.695i, **Z** = 1.694 + 0.901i, **V** = −5.351 + 2.733i.

## 6. Example

To demonstrate the kinematic synthesis solution procedure for a mechanism made up of dyads and triads, the MK and MKT circle methodology presented above is applied to a practical mechanism example. One common aesthetic and practical challenge in cabinets and boxes is how to implement the hinges to open the container lid. Perhaps the most used and inexpensive solution is a simple external single-axis hinge. However, external hinges have their drawbacks. They are difficult to conceal, potentially hampering the aesthetics of the container, and they take up space on the outside. Oftentimes kitchen cabinets will use external hinges, but the cabinet doors then need to be spaced far enough apart to allow space for these hinges. To solve this issue, internal hinges may be used. However, due to the thickness of the cabinet doors, single-axis hinges need to be inset into the cabinet door, or else they will be physically impossible to open. Moving beyond simple hinges, a linkage mechanism can resolve these challenges, as a mechanism allows for both translational and rotational motion. Here, a lid/door opening mechanism will be synthesized which first pushes the lid laterally away from the container, and then rotates the lid open, allowing the mechanism to reside fully inside the container.

**Solution**: first, a basic linkage topology must be selected. Based on observation and experience it is doubtful whether a basic four-bar linkage would be able to achieve the desired translation and rotation while still fitting inside a reasonably small space. The Watt 1 six-bar mechanism is known for producing complex motions, so that is the linkage topology selected. This mechanism is shown with its dyad and triad loops highlighted in Figure 8.

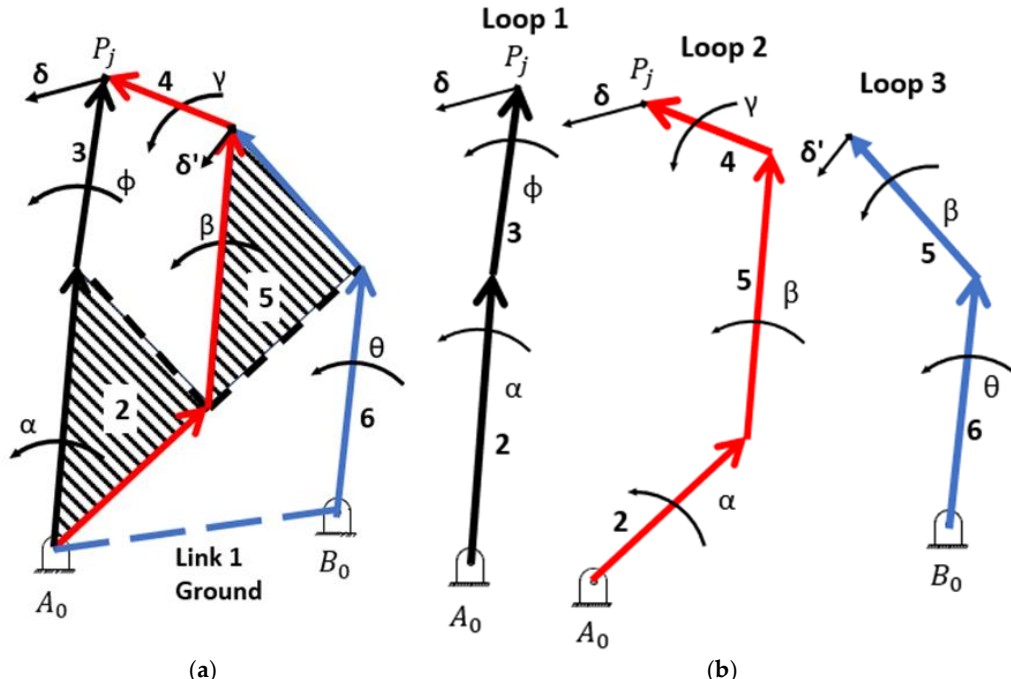

**Figure 8.** (**a**) The Watt 1 topology is shown, with the individual loops highlighted. (**b**) The loops are separated to show the individual synthesis chains more clearly. Loop one and loop three are dyads shown in black and blue, respectively, while loop two is a triad shown in red.

This problem will be solved using four prescribed positions of the motion generation link, here chosen as the third link in the triad. The first two positions will guide the translation of the lid away from the container, while the next two focus on rotating the lid away from the opening, as shown in Figure 9.

To synthesize the Watt 1 mechanism, first, the dyad identified as loop one in Figure 8b is synthesized. Then, the triad (loop two) is designed; note that the triad loop shares a ground pivot and the rotation angle β of its first link with the loop one dyad. Finally, the third loop (another dyad) is synthesized. This dyad's ground pivot location is completely free of the other two, but its second link shares its rotation angles with the second link of the triad. The key synthesis process details are summarized in Table 3.

**Table 3.** Summary of known and unknown quantities for Watt 1 topology.

| Loop # | Prescribed Values | Unknowns | Connection |
|--------|-------------------|----------|------------|
| Loop 1 | $\delta_{2-4}$ | $Oa$, $\beta_{2-4}$, $\alpha_{2-4}$ | - |
| Loop 2 | $\delta_{2-4}$, $\beta_{2-4}$, $Oa$ | $\alpha_{2-4}$ | $\beta_{2-4}$, $Oa$ from Loop 1 |
| Loop 3 | $\delta'_{2-4}$, $\alpha_{2-4}$ | $\beta_{2-4}$ | $\delta'_{2-4}$, $\alpha_{2-4}$ from Loop 2 * |

\* $\delta'_j$ is calculated from $PP_j$ minus $V_j$, as shown in Equation (9).

### 6.1. Loop One

The first dyad has additional free choices that are not afforded to the other loops. In the topology shown in Figure 8a, the displacement angles of the first link are the same as the first link of the triad. This means that both the $\beta_j$ and $\alpha_j$ angles are initially free selections for this dyad, but their values have great implications for the other two chains. Thus, even for four prescribed positions, there are many more possible solutions for this dyad than can be represented by a single selection of input parameters. In Figure 10, the MK circles for the selected four-precision position dyad problem are shown.

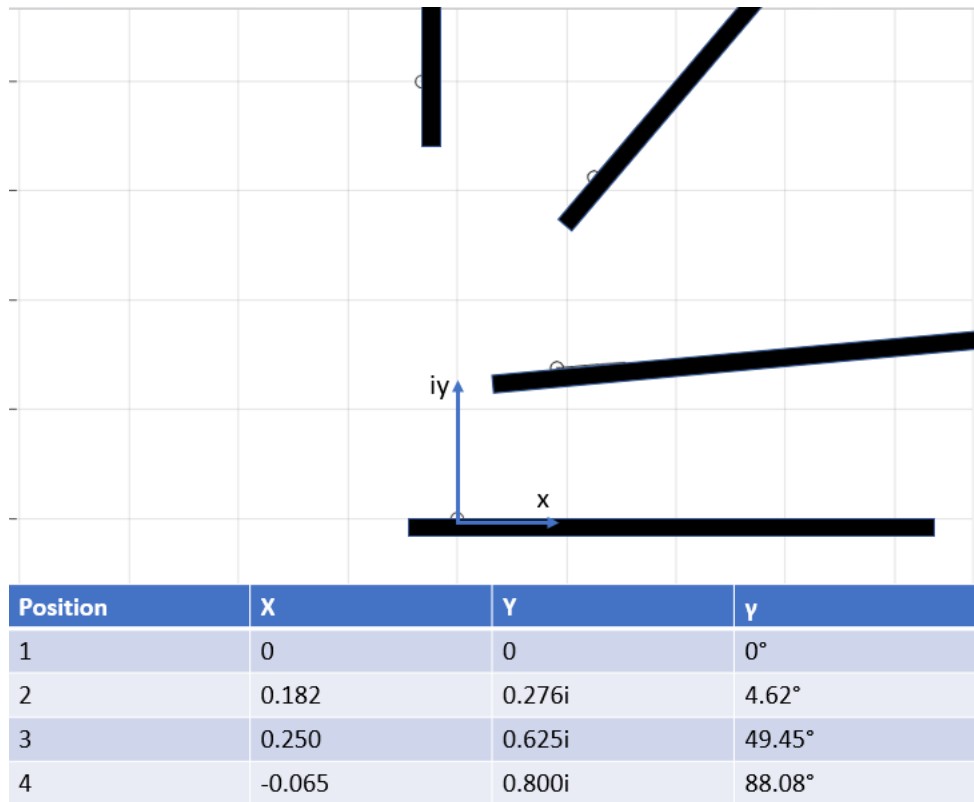

| Position | X | Y | γ |
|----------|-------|--------|--------|
| 1 | 0 | 0 | 0° |
| 2 | 0.182 | 0.276i | 4.62° |
| 3 | 0.250 | 0.625i | 49.45° |
| 4 | -0.065 | 0.800i | 88.08° |

**Figure 9.** The prescribed motion generation positions.

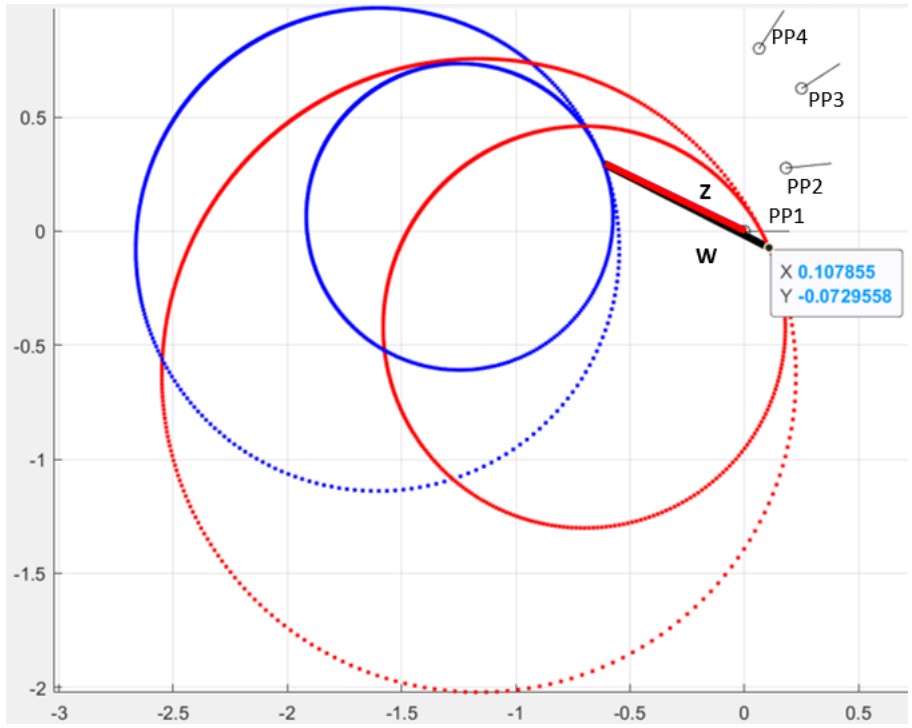

**Figure 10.** The MK circles for the loop one dyad of the final mechanism, with a dyad solution shown. W = 0.786 − 0.208i, and Z = −0.230 + 0.644i. The $\beta_j$ angles were set as $\beta_2 = -18.582$, $\beta_3 = -21.767$, $\beta_4 = -12.517$. The free choice was chosen as $\alpha_2 = 5.862$.

*6.2. Loop Two*

The synthesis procedure becomes a bit more complex with the introduction of the second synthesis loop, the triad. Here, to affirm the Watt 1 topology, the ground pivot location must be specified to match the dyad formed in loop one. Additionally, all the β and γ angle values are specified, as $\gamma_{1-4}$ are given in the problem definition, and $\beta_{1-4}$ must match the β values of the dyad calculated in loop one or else they cannot combine into the single ternary link 2 in Figure 8a. This means that only the α angles remain unprescribed. Any of these three $\alpha_j$ angles may be chosen as a free choice, but the other two must be solved for. One way to do this with the MKT circles is to iterate through many values of $\alpha_3$ and $\alpha_4$ for a given value of $\alpha_2$, then pick out the set of pivots based on whichever set produces a matching ground pivot to loop one. While inexact, designers should be able to quickly find matching solutions within four or five decimal point accuracy by increasing the number of selections considered for $\alpha_3$ and $\alpha_4$. This procedure is applied in Figure 11. Only the ground pivot circles are shown, with 50 circles plotted comprised of 200 points each.

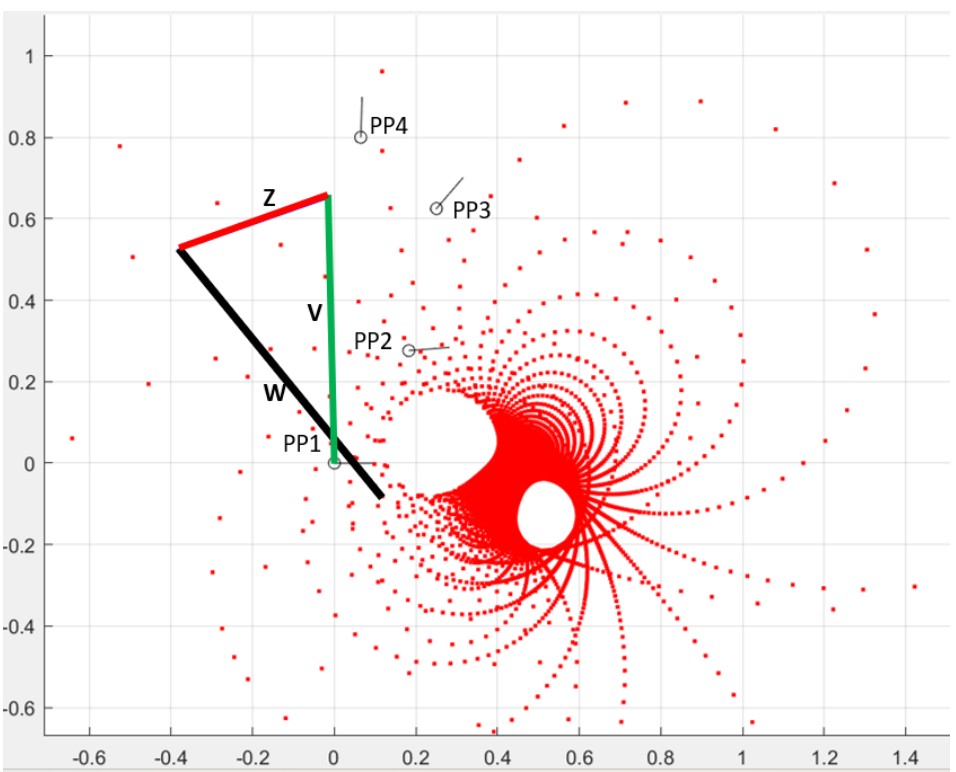

**Figure 11.** The ground pivot circles of a triad, plotting numerous values of $\alpha_3$ and $\alpha_4$. The $K_1$ and $T_1$ circles are hidden to simplify the figure. The matching ground pivot location is highlighted.

Many designers will find it informative to generate and examine Figure 11 prior to synthesizing the loop one dyad, due to its ability to reveal so-called 'forbidden regions' [1–4]. These are areas in the plane where no viable ground pivot solutions exist, despite a high solution density elsewhere. Figure 11 shows two of these pockets, with one centered around $(0.55, -0.1)$ and the other just above the selected ground pivot around $(0.3, 0.05)$. Varying the value of $\alpha_2$ will cause these regions to shift, but if a particular value of $\alpha_2$ is desired for the triad, it may be necessary to reassign the free choice values of the loop one dyad to make finding a viable solution possible.

Once again, each circle represents a unique angle value of $\alpha_3$, and each pivot point on the circles is plotted via a unique value of $\alpha_4$ (or vice versa). So, once the matching ground pivot point is identified, the values of $\alpha_3$ and $\alpha_4$ are inherently known—they are whichever two values were used to create that point. In this example, the solution is found when

$\alpha_3 = 90.062$ and $\alpha_4 = 132.141$ degrees, based on a free-choice value of $\alpha_2 = 24.919$ degrees. This value of $\alpha_2$ was selected after some trial and error with the final mechanism construction, as the value appeared to produce favorable motion results.

### 6.3. Loop Three

The second dyad may be synthesized in the same manner as the first dyad—finding the intersection of two sets of MK circles. Now that the triad chain has been identified, the $\alpha_j$ angles for the loop three dyad are prescribed, as they must match with the $\alpha_j$ values of the triad to form link five, shown in Figure 8. It is also important to note that the precision positions are no longer those specified in the first two loops, but rather the distal precision position minus the **V** vector of the triad at this position. This new distal displacement is symbolized by $\delta_j'$, determined as shown in Equation (16).

$$\delta_j' = \delta_j + \mathbf{V}\left(e^{i\gamma_j} - 1\right) \tag{16}$$

This is due to the chosen loop configuration for the Watt 1 topology shown in Figure 8, which requires that the second dyad extend up to the end of the second link of the triad. This also means that rather than using the $\gamma_j$ angles to define the motion, the $\alpha_j$ angles of the triad define the motion.

For this second dyad, there is no ground pivot specification required, meaning any candidate solutions that meet the overall design parameters will be sufficient. The problem can be quickly solved using the standard MK circle approach, as shown in Figure 12. To show that loop-based approaches to assembling multi-chain mechanisms are interchangeable, this second dyad was also synthesized using the previously developed compatibility linkage approach [5]; a sample calculation is shown in Appendix A.

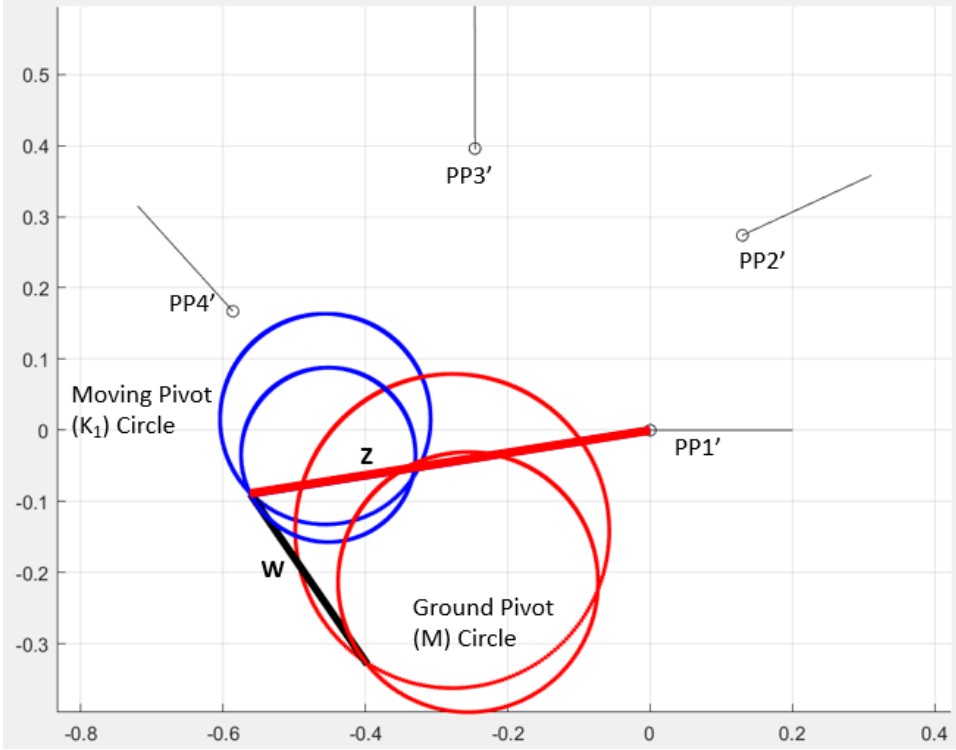

**Figure 12.** A depiction of the MK circles for the loop three dyad. PP1 = 0 + 0i, PP2 = 0.130 + 0.274i PP3 = −0.246 + 0.396i, PP4 = −0.586 + 0.167i. $\alpha_2$ = 24.919, $\alpha_3$ = 90.062, $\alpha_4$ = 132.141. Solution found for $\beta_2$ = −45.508, $\beta_3$ = −90.331, $\beta_4$ = −95.758.

All the required synthesis information for the Watt 1 mechanism is now compiled. The final step is to combine the necessary segments from different loops and assemble the completed mechanism. It is important to note that this solution procedure does not have built-in checks for circuit, branch, or other types of motion defects that could prevent a mechanism from operating as intended. Identifying and eliminating these defects will require some additional work after the initial synthesis process has been completed. Finding and eliminating defects has been a field of extensive study in kinematic synthesis [19–24]. Balli and Chand wrote an effective review of these topics which can be found in reference [25]. An image of the completed mechanism in its four prescribed positions is shown in Figure 13. For this prototype, the mechanism was scaled up by a factor of 10 to increase the ease of prototype construction and to make the motion more visible. The key unscaled linkage parameters are summarized in Table 4.

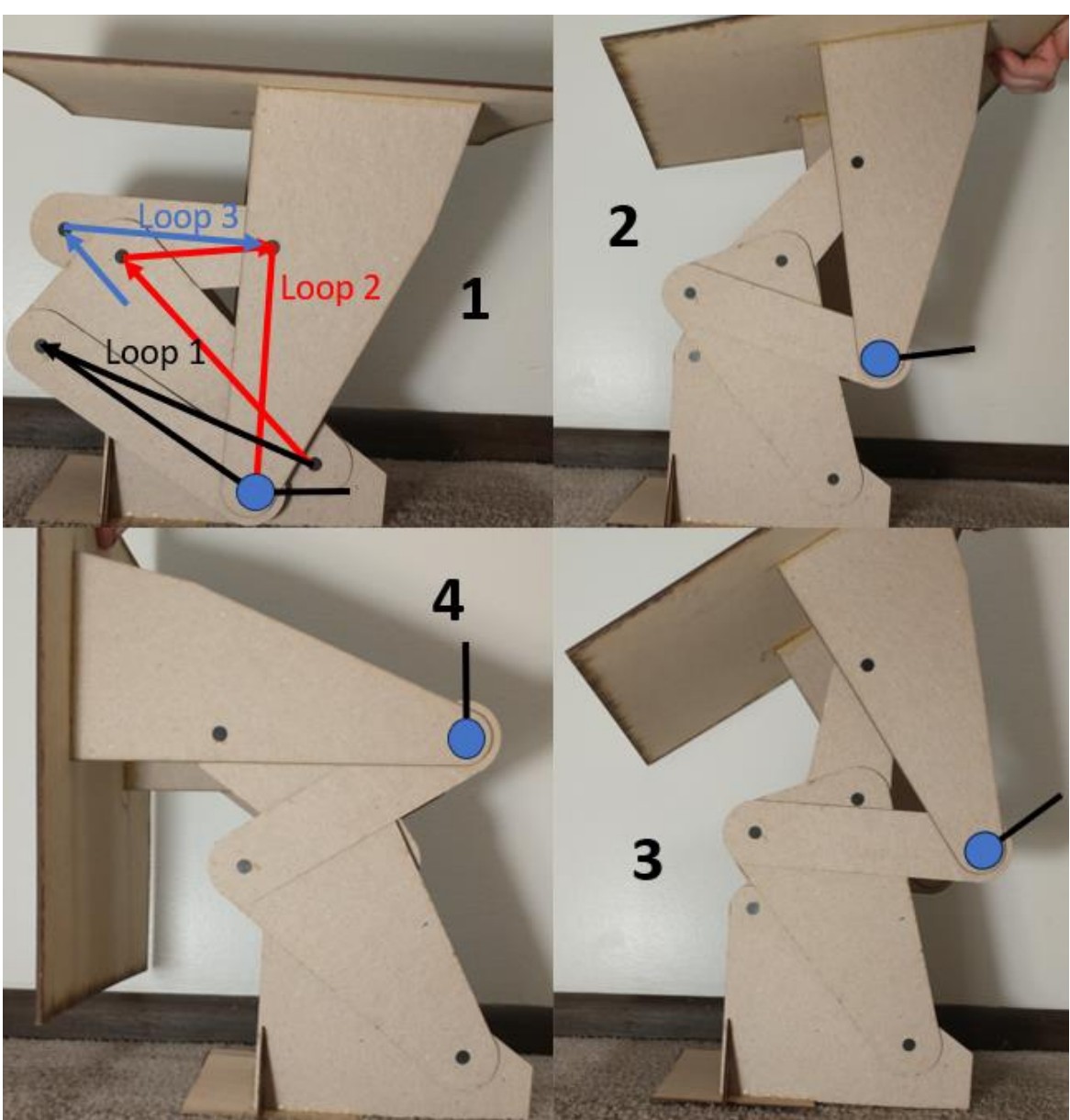

**Figure 13.** An assembled prototype visualized at a few points in its motion. The loops are labeled in the first position. Clockwise from top left, positions one to four.

**Table 4.** Solution vector chains for each loop.

| Loop | W [1] | Z | V |
|:---:|:---:|:---:|:---:|
| 1 | $-0.725 + 0.369i$ | $0.617 - 0.296i$ | - |
| 2 | $-0.504 + 0.603i$ | $0.369 + 0.129i$ | $0.017 - 0.651i$ |
| 3 | $-0.164 + 0.239i$ | $0.562 + 0.090i$ | - |

[1] All values are for the mechanism in its first position and rounded to the third decimal point.

## 7. Conclusions

The methods presented in this paper, including advancement in MK circle usage, and the expansion to MKT circles for triads, is demonstrably an effective technique for creating dyad and triad chains, as well as for gaining information about the solution space for pivot locations for planar synthesis problems. Triad MKT circles are shown to be useful for achieving ground pivot specification, a practice which is often essential for multi-loop synthesis. Finally, a geometry-based approach to identifying circle intersection points with respect to the free choice variable is shown, making it possible to identify solutions rapidly and precisely. The discussed techniques are used to synthesize a Watt 1 mechanism by dividing the mechanism into three distinct loops which may be solved independently. Additionally, the present method is shown to be interchangeable with the compatibility linkage method, another technique for multi-loop mechanism synthesis. New understandings have been introduced for extending the MK circles solution method beyond three prescribed positions for motion generation.

There remain opportunities to develop these methods further. In triad synthesis, our findings in Figure 6 seem to indicate that an equivalent structure to the pole observed in dyads may exist for triads as well. A few special cases are known, e.g., $\beta j = \alpha j$, but the majority of these potential pole locations are left for future work. Additionally, the theory should be equally applicable to solving quadriad chains; after reworking the standard form equations, the rest of the procedure should follow in the same fashion as the dyad and triad forms.

**Author Contributions:** Conceptualization, S.M. and A.E.; methodology, S.M. and A.E.; software, S.M.; validation, S.M. and A.E.; formal analysis, S.M. and A.E.; investigation, S.M. and A.E.; resources, S.M. and A.E.; data curation, S.M.; writing—original draft preparation, S.M. and A.E.; writing—review and editing, S.M. and A.E.; visualization, S.M. and A.E.; supervision, A.E.; project administration, A.E.; funding acquisition, None. All authors have read and agreed to the published version of the manuscript.

**Funding:** This research received no external funding.

**Data Availability Statement:** No new experimental or research data created for this work.

**Conflicts of Interest:** The funders had no role in the design of the study; in the collection, analyses, or interpretation of data; in the writing of the manuscript; or in the decision to publish the results.

## Appendix A

Here, a numerical example of the compatibility linkage method is shown to demonstrate that the MK/MKT circles and other loop-based synthesis methodologies are interchangeable. The standard form equations for a dyad in four positions are shown in Equation (A1).

$$\begin{bmatrix} e^{i\beta_2} - 1 & e^{i\alpha_2} - 1 \\ e^{i\beta_3} - 1 & e^{i\alpha_3} - 1 \\ e^{i\beta_4} - 1 & e^{i\alpha_4} - 1 \end{bmatrix} \begin{bmatrix} \mathbf{W} \\ \mathbf{Z} \end{bmatrix} = \begin{bmatrix} \delta_2 \\ \delta_3 \\ \delta_4 \end{bmatrix} \tag{A1}$$

The parameters used to define this problem are identical to those laid out for the loop three dyad in the example in the main text. They are summarized in Table A1.

**Table A1.** Summary of known and unknown quantities for Watt 1 topology.

| Position # | Position Coordinates | $\beta$ | $\alpha$ |
|:---:|:---:|:---:|:---:|
| 1 | 0 + 0i [1] | 0 | 0 |
| 2 | 0.130 + 0.274i | $-44.775°$ [2] | 24.919° |
| 3 | $-0.246 + 0.396i$ | Unknown | 90.062° |
| 4 | $-0.586 + 0.167i$ | Unknown | 132.141° |

[1] The position coordinates are uniformly adjusted to have PP1 as (0, 0). [2] The value of the free choice $\beta_2$ used in the MK solution for loop three was $-45.508$. The value was modified by $0.733°$ for this example to account for rounding errors within the solution software and to produce an identical dyad solution.

The solution procedure is only briefly described here. Consider references [2,5], for more information on how the compatibility linkage method is derived and its many applications. Equations (A2)–(A10) show the general form of the key equations used for a dyad in four prescribed positions. From the problem definition, all the values in (A2)–(A4) are known, and in (A6) $\beta_2$ is a free choice.

$$\Delta_2 = \begin{vmatrix} e^{i\alpha_3} - 1 & \delta_3 \\ e^{i\alpha_4} - 1 & \delta_4 \end{vmatrix} \tag{A2}$$

$$\Delta_3 = -\begin{vmatrix} e^{i\alpha_2} - 1 & \delta_2 \\ e^{i\alpha_4} - 1 & \delta_4 \end{vmatrix} \tag{A3}$$

$$\Delta_4 = \begin{vmatrix} e^{i\alpha_2} - 1 & \delta_2 \\ e^{i\alpha_3} - 1 & \delta_3 \end{vmatrix} \tag{A4}$$

$$\Delta_1 = -\Delta_2 - \Delta_3 - \Delta_4 \tag{A5}$$

$$\Delta = \Delta_1 + \Delta_2 e^{i\beta_2} \tag{A6}$$

By the law of cosines,

$$x = \frac{|\Delta_4|^2 - |\Delta_3|^2 - |\Delta|^2}{2|\Delta_3||\Delta|} \tag{A7}$$

$$\beta_3 = angle(\Delta) + \text{acos}(x) - angle(\Delta_3) \tag{A8}$$

Note that for the angle of $\beta_3$ shown in Equation (A8), and for $\beta_4$ in Equation (A10), it is required that $|x_j| < 1$. If x is outside this range, it is outside the domain of inverse cosine, and therefore will produce complex answers when a scaler angle is desired. In this case, a special procedure using the atan2(x,y) function is implemented. See reference [2] (p. 182) for more details. The $\beta_4$ angles are evaluated in the same manner as $\beta_3$, with slight modifications to accommodate the distinct geometry.

$$x_2 = \frac{|\Delta_3|^2 - |\Delta_4|^2 - |\Delta|^2}{2|\Delta_4||\Delta|} \tag{A9}$$

$$\beta_4 = angle(\Delta) + \text{acos}(x_2) - angle(\Delta_4) \tag{A10}$$

Evaluating Equations (A2)–(A10) using the values provided in Table A1 results in $\beta_3 = 269.067$, and $\beta_4 = -96.572$. Either of these values may be chosen and plugged into Equation (1) to solve for the values of **W** and **Z**. The compatibility linkage method ensures that the values of $\beta_3$ and $\beta_4$ will be compatible with all four positions, not just the first three. The resultant dyad is shown in its four positions in Figure A1 and is identical to the loop three dyad shown in Figures 12 and 13.

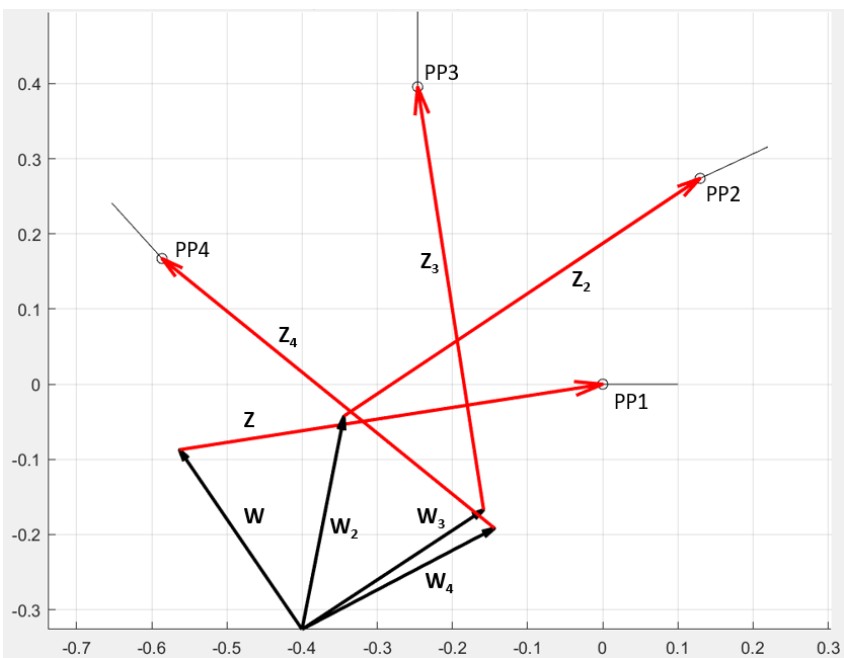

**Figure A1.** The resultant dyad synthesized using the compatibility linkage approach, matching the dyad found in Figures 12 and 13.

**Appendix B**

Figure A2a depicts a dyad, while Figure A2b depicts a triad in two prescribed positions. These constructions yield the standard form equations shown in Equations (1) and (8).

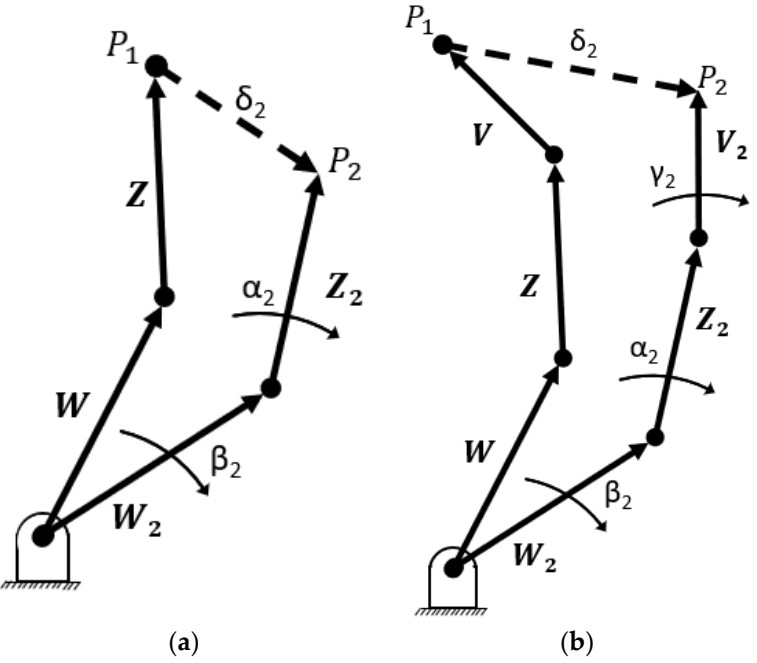

(a)                                      (b)

**Figure A2.** (**a**). A dyad shown in two prescribed positions, with key vectors and angular displacements labeled. (**b**). A triad shown in two prescribed positions, with key vectors and angular displacements labeled. Note that this triad notation assigns the vector link "V" to the third link of the triad chain, as opposed to the intermediate link where it has classically been assigned. We feel this updated nomenclature is more intuitive for designers who are well acquainted with dyads and hope to incorporate triads into their designs [5,26,27].

### Five Prescribed Position Synthesis for Dyads:

The strategy using MK circles to find possible dyad solutions for five positions continues from four position synthesis with the addition of one more set of circles. Typically, this additional circle set will represent positions one, two, and five. For a solution to exist, all three of these circle sets (123, 124, and 125) must intersect at a single point. While there can be zero, one, or even two such triple intersections, these solutions are few and far between. There are zero free choices in this case, as shown in Table 1. Therefore, rather than choosing the value of $\beta_2$, the quest is for a value of $\beta_2$ that yields a motion generation dyad for all five positions. This requires a nonlinear solution or an optional search [2,5]. Figure A3 shows one triple intersection and the resulting dyad.

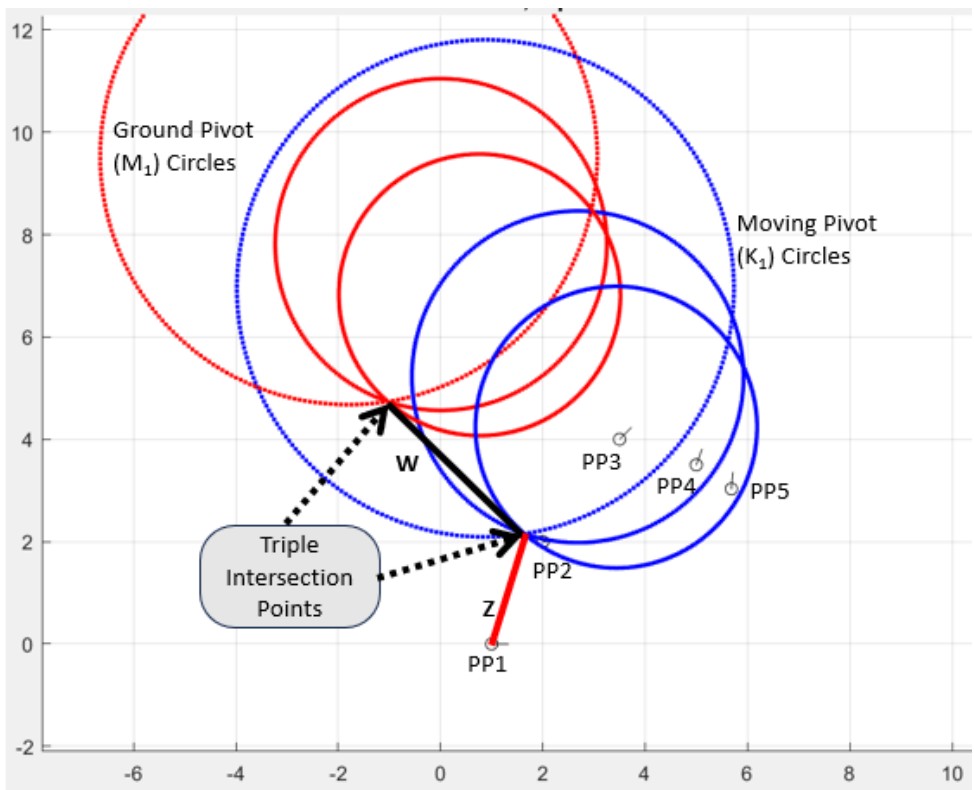

**Figure A3.** A depiction of a triple intersection point for the MK circles of a dyad in five positions. PP1 = 0 + 0.0i, PP2 = 1 + 2i, PP3 = 2.5 + 4i, PP4 = 4 + 3.5i, and PP5 = 4.6865 + 3.0306i. Note that when a solution point exists on the K circle, a matching solution will exist on the M circle as well. Here, **W** = 2.67 − 2.59i, and **Z** = −0.64 − 2.19i, $\alpha_2 = 0$, $\alpha_3 = 45$, $\alpha_4 = 70$, and $\alpha_5 = 85$. The solution is found when $\beta_2 = 35.002$. It is possible but not guaranteed that other values of $\beta_2$ may also produce solutions.

Recall that each of the points on the circles represents a unique solution to the motion problem. Reducing the number of plotted points and revealing the vectors in between these circles makes it much more apparent how each of the points on these circles represents a complete triad solution to the problem, as seen in Figure A4, which stems from the same problem definition shown in Figure 5.

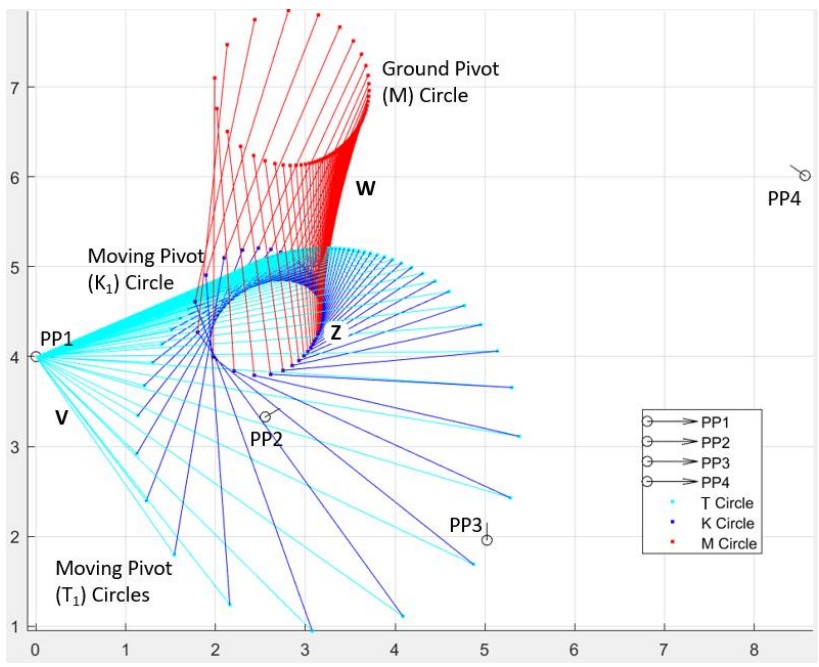

**Figure A4.** This Figure depicts the same circles as Figure 5, but here the vectors spanning between the circles are shown, demonstrating how each point on the circles represents a unique triad chain. PP1 = 0 + 4i, PP2 = 2.553 + 3.329i, PP3 = 5.023 + 1.957i, PP4 = 8.564 + 6.014i. $\beta_2 = 10$, $\beta_3 = 70$, $\beta_4 = 140$, $\alpha_2 = 60$, $\gamma_2 = 30$, $\gamma_3 = 90$, $\gamma_4 = 145$.

**Six Position Case:**

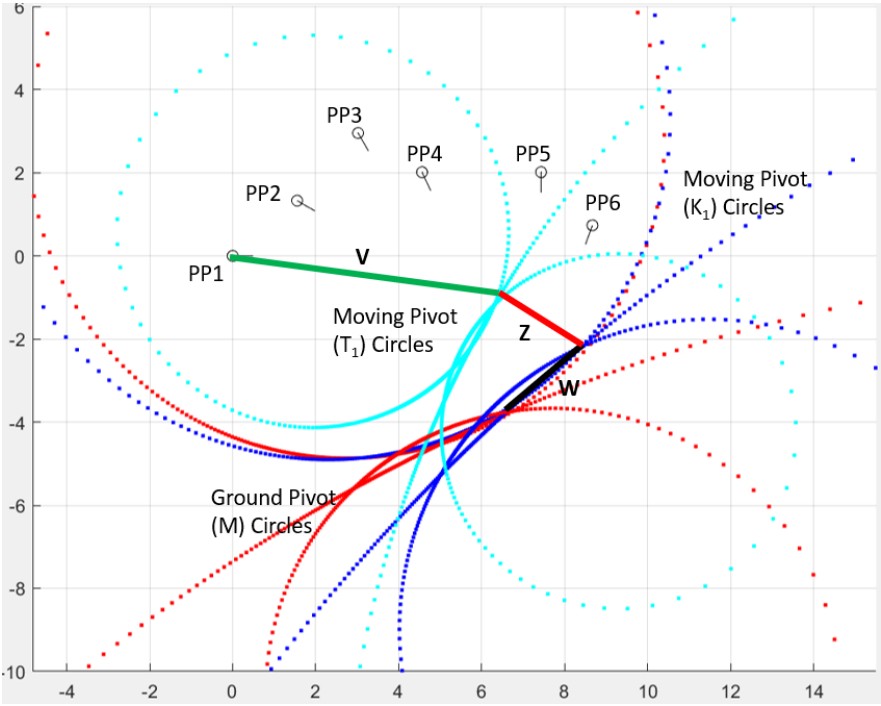

**Figure A5.** The triad MKT circles are shown in six prescribed positions, with one possible solution triad highlighted. The solution requires three circles intersecting at a single point to find solutions. PP1 = 0 + 0.0i, PP2 = 1.553 + 1.329i, PP3 = 3.023 + 2.957i, PP4 = 4.564 + 2.014i, PP5 = 7.432 + 2.016i, PP6 = 8.670 + 0.737, $\beta_2 = 30$, $\beta_3 = 70$, $\beta_4 = 140$, $\beta_5 = 150$, $\beta_6 = 130$, $\gamma_2 = -30$, $\gamma_3 = -60$, $\gamma_4 = -65$, $\gamma_5 = -90$, $\gamma_6 = -110$. A solution vector is found for $\alpha_2 = -15$. **W** = 1.651 + 1.567i, **Z** = −2.134 + 0.998i, **V** = −6.326 + 1.137i.

**Seven Position Case:**

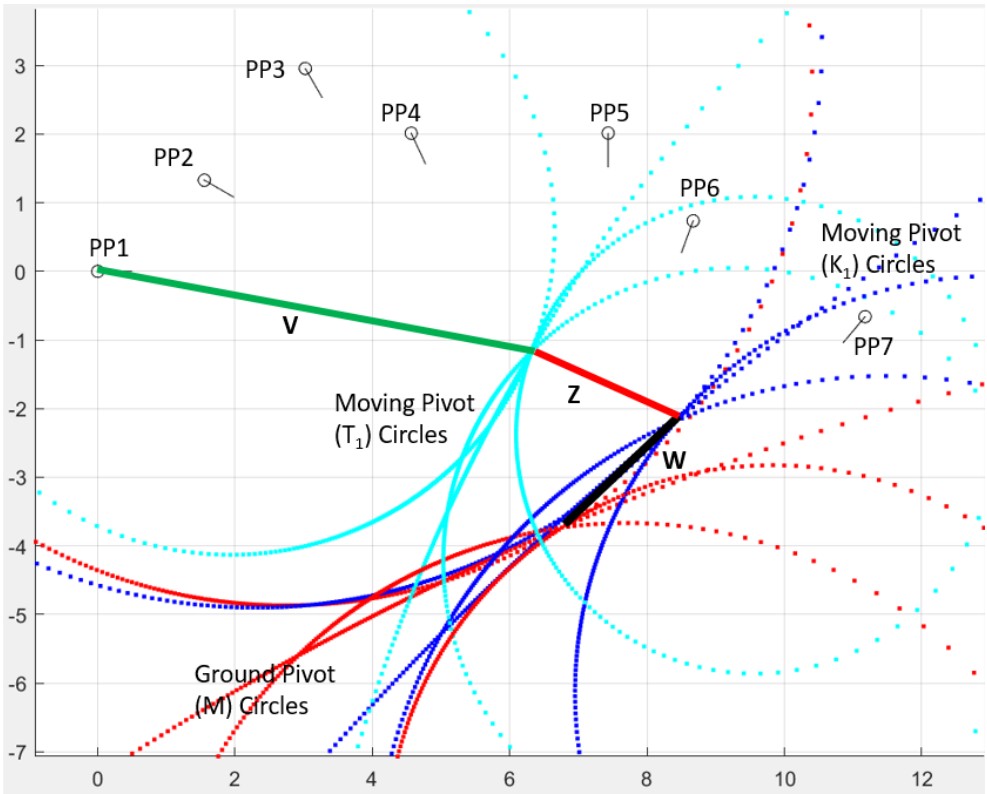

**Figure A6.** The triad MKT circles are shown in seven prescribed positions, with one possible solution triad highlighted. PP1 = 0 + 0.0i, PP2 = 1.553 + 1.329i, PP3 = 3.023 + 2.957i, PP4 = 4.564 + 2.014i, PP5 = 7.432 + 2.016i, PP6 = 8.670 + 0.737, PP7 = 11.174 − 0.659i, $\beta_2$ = 30, $\beta_3$ = 70, $\beta_4$ = 140, $\beta_5$ = 150, $\beta_6$ = 130, $\beta_7$ = 160, $\gamma_2$ = −30, $\gamma_3$ = −60, $\gamma_4$ = −65, $\gamma_5$ = −90, $\gamma_6$ = −110, $\gamma_7$ = −130. A solution vector is found for $\alpha_2$ = −15. **W** = 1.651 + 1.567i, **Z** = −2.134 + 0.998i, **V** = −6.326 + 1.137i.

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
