# Peer review of "Expansion of MK Circle Theory for Dyads and Triads"

_machines, doi:10.3390/machines11080841_

Round 1
Reviewer 1 Report
The manuscript is generally very interesting. However, it needs some revisions before it can be accepted for publication.
Please improve the introduction.
Different fonts used in 1. Introduction and the rest of the paper (please double-check)
The fonts used in Equation 1 are larger than in the other equations.
In Figure 3, the poles P23’, P23, and P13 are placed in a kind of box with white background, which makes the paths/trajectories look like they are interrupted.
Row 106 is missing a space between P13 and is
Figure 4, the same issue as in Figure 3.
Before Figures 5, 6, 7, 9, 10, 11, 13, 15, 16 is too much space.
Rows 199-206 – the font seems very small.
Row 239: change 6PP triad with PP6 triad
Rows 343-347 – the font seems very small compared to the above and the following paragraphs.
Row 377 appears to be empty.
Between rows 377 and 378 there is a large blank space that must be deleted
In Appendix A, equation A1 uses larger fonts compared with the rest of the equations in this appendix.
Delete the space above row 424
Delete the space above row 469
Review the references’ formatting.
Author Response
Thank you for your thoughtful feedback! We have been able to complete most of your requested formatting changes. We appreciate you pointing out each of those needed revisions. We have had difficulty in fully eliminating a few of the blank spaces that you noticed due to the large number of figures in the document. They make the spacing of the document rather tricky, as there was not always enough text in a given section to fill the space between the figures.
We have also tried to make improvements to the introduction by clarifying what sections of the document are new and what sections have been covered in previous work. If you have specific recommendations for additional improvements to the introduction, we are open to hearing them!
Reviewer 2 Report
11. It is well-known that complex numbers cannot be used in the analysis and synthesis of three-dimensional mechanisms. Moreover, not many readers are familiar with complex numbers. Therefore, the use of vectors is preferred. However, the authors of the manuscript under review use complex numbers. This could be considered a disadvantage of the method proposed by the authors.
22. References [4], [6], and [11] are incomplete.
33. Ultimately, a dyad or a triad is an assembly of links and joints. Therefore, it is important to indicate what type of joints are involved. However, in Figures 1 and 6, the authors do not explicitly indicate the type of joint that exists between the corresponding links. Moreover, each position vector arises from two different points, which do not appear in these figures.
44. In general, the quality of the figures shown is not very good. Particularly Figures 1 and 6 seem to me to have been seen in articles and books written many years ago. It would be strongly recommendable for the authors to improve the quality of the figures.
55. Graphic constructions predominate throughout the article. The inclusion of so many graphic constructions affects the readability of the article and distracts the reader's attention. In addition, this makes the method very laborious and difficult to follow.
66. Designers usually seek the highest number of precision points by using the simplest mechanisms, i.e., those with the fewest number of joints and links. However, the authors use a Watt mechanism to achieve only four points of accuracy (poses). On the other hand, it is well-known that a planar four-bar 4R-type mechanism can achieve up to five given poses. So, if a planar 4R mechanism is simpler than a Watt mechanism, why use a Watt mechanism?
77. If one considers the organization of the sections and subsections presented by the authors, it can be noted that there is a considerable amount of reading to be done and in the end, the reward is very little. Only one practical example is shown.
88. Without the slightest intention to offend and in the most respectful manner possible, it seems to me that the manuscript under review is no more than a somewhat disorganized compilation of the results reported in references [1]-[6], and [11]-[13]. In particular, given the topic covered in reference [3], it makes the contribution of the manuscript under review almost null.
99 Honestly, and in the most respectful way possible, I recommend the authors completely reformulate the manuscript, eliminating everything superfluous and what has already been dealt with several times in preliminary works, going straight to the point and explicitly indicating those contributions that are novel and original. The sections and subsections must be organized efficiently in such a way as to guide a standard reader through the shortest path, avoiding superfluous explanations.
Author Response
Thank you for your feedback! We appreciate the reviewer’s intentionality in delivering negative feedback in a polite and constructive way. We have been able to make many of the changes you recommended, and we hope that the revised version of the manuscript is able to address many of your concerns! Each of your comments is quite well-structured and distinct, so we’ll respond to them point by point.
- It is well-known that complex numbers cannot be used in the analysis and synthesis of three-dimensional mechanisms. Moreover, not many readers are familiar with complex numbers. Therefore, the use of vectors is preferred. However, the authors of the manuscript under review use complex numbers. This could be considered a disadvantage of the method proposed by the authors.
We agree complex numbers are not conducive to solving three-dimensional mechanisms. However, in the case of planar mechanisms, as are considered in the present paper, we actually feel that complex number-based methods provide many advantages over other approaches. Ultimately, this is largely a matter of preference, and there remains a sizable contingent of researchers who continue to teach and perform their research using complex numbers.
- References [4], [6], and [11] are incomplete.
Thank you for pointing this out! We have updated the references and added some additional ones to the paper.
- Ultimately, a dyad or a triad is an assembly of links and joints. Therefore, it is important to indicate what type of joints are involved. However, in Figures 1 and 6, the authors do not explicitly indicate the type of joint that exists between the corresponding links. Moreover, each position vector arises from two different points, which do not appear in these figures.
This is a helpful observation. While the method of MK circles is applicable to both revolute and prismatic joints, in this paper we are primarily focused on pinned joints. As such, we have added a statement to the introduction (lines 45-49) clarifying that all figures in the document display pinned joint solutions. To the second half of the comment, we agree that each delta position vector arises from two different points. However, perhaps there was a misunderstanding in interpreting the figures. Figures 1 and 6 depict a dyad and a triad, respectively, in two prescribed positions. The delta vector, labeled in the figures, spans the gap between the first and second position.
- In general, the quality of the figures shown is not very good. Particularly Figures 1 and 6 seem to me to have been seen in articles and books written many years ago. It would be strongly recommendable for the authors to improve the quality of the figures.
We apologize that you are not impressed with the quality of the figures! If you have any specific recommendations as to how they might be improved, we are open to hearing them. Special attention has been paid to improving the legibility of Figures 1 and 6 (these Figures have also been moved to Appendix B). As you observed, these figures are reminiscent of figures that have existed in the literature for decades. However, we feel the reason they keep coming up is that it is essential to establish the nomenclature for how vectors and angles will be labeled, as many kinematics papers use their own unique naming conventions.
- Graphic constructions predominate throughout the article. The inclusion of so many graphic constructions affects the readability of the article and distracts the reader's attention. In addition, this makes the method very laborious and difficult to follow.
As for the issue of graphic constructions oversaturating the article, we can certainly see why you felt this was an issue, there are a lot of graphs. We were hesitant to cut many of the figures, though, as part of our goal in writing this paper was to compose a thorough collection of the existing information on MK circles so that it could be all together in one resource. To address the concern, we have moved several of the figures, including Figures 1 and 6, out of the body of the text and into Appendix B. We hope that this adjustment makes the core points of the article clearer and more digestible without sacrificing some of the key information.
- Designers usually seek the highest number of precision points by using the simplest mechanisms, i.e., those with the fewest number of joints and links. However, the authors use a Watt mechanism to achieve only four points of accuracy (poses). On the other hand, it is well-known that a planar four-bar 4R-type mechanism can achieve up to five given poses. So, if a planar 4R mechanism is simpler than a Watt mechanism, why use a Watt mechanism?
This comment we disagree with. In our experience, designers typically try to solve problems with three to four precision positions (for dyads), as increasing the number of precision positions further makes it so that the designer no longer has free choices in designing the mechanism. We have found that many prefer to have a larger solution space to explore rather than prescribing more positions. Most of the time, 5 positions for the dyad and 7 positions for the triad would only be used in a case where a mechanism absolutely required all those prescribed positions. In terms of choosing the Watt mechanism, we agree that designers prefer to choose topologies that have the least number of links and joints when possible. However, the key point is achievable complexity of motion. Simple four-bar mechanisms are not able to reproduce the complex motions achievable by the Watt mechanism. We try to address this idea in lines 306-313, where we argue that a four-bar solution would not meet other constraints of the problem, like having a small profile, finding solutions with pivots in acceptable positions, or translating the lid away from the container and then rotating it open in consecutive steps.
- If one considers the organization of the sections and subsections presented by the authors, it can be noted that there is a considerable amount of reading to be done and in the end, the reward is very little. Only one practical example is shown.
While there is only one practical example shown in the document, we feel the example is quite comprehensive. It covers each of the three loops required in a Watt one mechanism, demonstrating the effectiveness of the MK solution methodology and the new approaches introduced in the paper. It is also (to the author’s knowledge) the first comprehensive example that demonstrates the compatibility of different loop-based synthesis methods, including the MK circle approach and the compatibility linkage method shown in Appendix A. There are also concerns about space—this example, because we intended to be thorough, spans 6 pages in the manuscript. Relating to the reviewer’s other points in comments 5 and 9, we feel adding an additional example would be excessive.
8-9. Without the slightest intention to offend and in the most respectful manner possible, it seems to me that the manuscript under review is no more than a somewhat disorganized compilation of the results reported in references [1]-[6], and [11]-[13]. In particular, given the topic covered in reference [3], it makes the contribution of the manuscript under review almost null.
Honestly, and in the most respectful way possible, I recommend the authors completely reformulate the manuscript, eliminating everything superfluous and what has already been dealt with several times in preliminary works, going straight to the point and explicitly indicating those contributions that are novel and original. The sections and subsections must be organized efficiently in such a way as to guide a standard reader through the shortest path, avoiding superfluous explanations.
Thank you again for the tactful and respectful delivery of your review. It is understandable that you felt this way, as after reading your feedback and reviewing the manuscript ourselves, we agreed that the specific conclusions of the article were not completely clear. With that said, we do feel that the article makes several new contributions, and we have done our best to clarify those contributions in the abstract, introduction, and conclusion. Additionally, much of the content that you may have felt was superfluous (based on your comment #5), has been moved to an Appendix to clear the way for the key ideas.
The main contributions we feel the manuscript makes are the following:
First, as you acknowledge in a few of your comments, several of the sections in this paper, especially relating to dyads, has been covered in previous literature. However, we do not feel that this is a weakness, but rather a strength of the paper. This manuscript compiles the existing information on the method of MK circles in a way that has not been done before, and we find it useful to have all the key information in one place.
Second, by demonstrating how the MK circle method is applied for multi-loop mechanisms (no previous work demonstrates how to combine dyads and triads through this method), we expand on the existing literature and make the topic more understandable/usable for the reader. Concepts like “forbidden regions” which were the primary focus of Mlinar’s paper are much clearer in the context of a real problem. The practical example also demonstrates triad ground pivot specification, a topic which (to the author’s knowledge) has not been specifically addressed in any kinematic synthesis paper. Finally, the example shows that different synthesis methods are interchangeable, a valuable point in understanding broader kinematic synthesis concepts which has not previously been addressed.
Third, this paper applies for the first time the MKT circle concept to triad synthesis in 5, 6, and 7 prescribed positions. While Mlinar did introduce the concept of triad MK circles, he did not go beyond four prescribed positions, and as mentioned above, his work focused almost entirely on the existence of forbidden regions in the triad case. We demonstrate how forbidden regions might be applied to a real example, and we increase the complexity of his work by showing how MK circles may still be applied in problems with more prescribed positions.
Fourth, and finally, we added an additional section to the paper clarifying how circle intersections are found in these higher position cases. This algorithm is a logical expansion from the geometry of the MK circles relating to the poles. However, the math is not trivial, and by introducing a specific procedure for finding intersections, it becomes much more accessible to rapidly find all β2 solutions to a given problem, even in the five-position case. Hopefully, the additions in the introduction and conclusion help clarify these aims and distinguish the new and old material.
Reviewer 3 Report
As a main comment, it is worth noting the lack of a clearly defined scientific novelty. The authors write in the abstract about the new theory and methodology, but then there is no section where some new method is described that is fundamentally different from the existing ones. The authors refer to other works where the MK Circle Theory is applied, including for triads, but do not substantiate the novelty of the fundamental contribution of this work.
Author Response
Thank you for your feedback! Your main comment was that the novelty of the paper was not clearly defined at any point. To this end, we have added some sentences to the abstract, introduction, and conclusion which more clearly define what we feel the contribution of the paper is. Additionally, we have moved some of the content from the body of the text to an appendix. Our hope is that isolating the key information in the body of the text will make the contributions clearer and make the work more digestible and useful for the reader.